# Association of remittances with skilled delivery in Uganda, 2019/2020

**Chelsea Ducille**[1,2]*, **Richard Bilsborrow**[2,3◐], **Kavita Singh**[1,2◐], **Jon Hussey**[1,2‡], **Larissa Jennings Mayo-Wilson**[1,2,4‡]

**1** Department of Maternal and Child Health, University of North Carolina at Chapel Hill, Chapel Hill, North Carolina, United States of America, **2** Carolina Population Center, University of North Carolina at Chapel Hill, Chapel Hill, North Carolina, United States of America, **3** Department of Geography and Environment, University of North Carolina at Chapel Hill, Chapel Hill, North Carolina, United States of America, **4** Department of Health Behavior, University of North Carolina at Chapel Hill, Chapel Hill, North Carolina, United States of America.

◐ These authors contributed equally to this work.
‡ JH and LJM-W also contributed equally to this work.
* cmducille@gmail.com

## Abstract

There are significant disparities in maternal and child mortality ratios between high-income and low- and middle-income countries (LMICs). Despite large decreases in maternal and child mortality in recent decades, with programs such as the Safe Motherhood Initiative, the Sustainable Development Goals and the Millennium Development Goals in LMICs, these differences remain large. The objective of this study is to examine the association between the receipt of remittances received by households and skilled delivery among women reporting a recent birth. This study uses data from the household, woman, and community modules of the Uganda National Panel Survey (UNPS) from 2019/2020 to examine skilled delivery and skilled delivery with the use of safe birthing kits. Women of reproductive age with a birth within the last three years of data collection were included in the analysis. A cross-sectional analysis of 2019/2020 UNPS data is analyzed using logistic regression and a dose-response analysis plotting the estimated probabilities of skilled delivery and skilled delivery with use of safe birthing kits against increasing amounts of remittances. While no statistically significant associations were found between remittances and skilled delivery in the logistic model, the dose-response model did find such a relationship. Recommendations for future research in this topic area include improving questionnaire design to collect better data on remittances, utilizing longitudinal data with larger sample sizes, and examining mechanisms of money transfer.

**Data availability statement:** Data used in this study is publicly accessible through the World Bank Microdata Library, central data catalog. It is part of the Living Standards Measurement Surveys sub-category. Data can be retrieved at https://microdata.worldbank.org/index.php/catalog/3902/get-microdata.

**Funding:** The author(s) received no specific funding for this work.

**Competing interests:** The authors have declared that no competing interests exist.

## Introduction

There are significant differences in maternal mortality ratios between high-income and low and middle-income countries (LMICs). Despite large decreases in maternal deaths over several decades, with programs like the Safe Motherhood Initiative, the Sustainable Development Goals and the Millennium Development Goals, maternal mortality rates remain the highest in LMICs [1–3]. Ninety-four percent of global maternal deaths occur in low and lower middle-income countries. Regionally, 86% of those deaths occur in sub-Saharan Africa and in South and Southeast Asia [2]. For example, over the last two decades, maternal health outcomes have improved in Uganda. Between 2001 and 2010 the country enacted several public health policies focused on improving healthcare access and reducing maternal and neonatal mortality. These policies included health center reform and capacity building, the reduction of healthcare costs, and prohibition of care from unskilled providers [4–6]. The maternal mortality ratio in Uganda continues on a downward trajectory. The global maternal mortality ratio is 223 deaths per 100,000 live births, compared with 536 deaths per 100,000 live births in Sub-Saharan Africa [7]. Despite 95% of women having at least one antenatal visit and 74% of births attended by skilled health staff, the lifetime risk of maternal mortality is 1 in 49 compared to 1 in 5,300 in high-income countries, illustrating the huge global disparities in maternal mortality [2,5,8,9].

Skilled births or skilled deliveries refer to births attended by a skilled health professional with education and training in maternal and newborn health. These health professionals usually refer to doctors (obstetricians, pediatricians, anesthesiologists, etc.), trained midwives, and nurses. As a maternal and newborn care team, these health professionals have the capabilities of providing emergency care to women and newborns. The World Health Organization's (WHO) definition of a skilled delivery also implies that trained health professionals perform their care in safe and clean environments [9]. Essential obstetric services are associated with a 74% decrease in maternal mortality in Sub-Saharan Africa [10]. Generally, as the level of obstetric care increases, maternal morality ratios decrease [11]. Though estimates vary, skilled delivery has been found to prevent between 16% to 33% of maternal and newborn mortality [12].

In Uganda, the health care system is largely decentralized into health subdistricts that are intended to be independent bodies financially supported by their own revenues. However, many subdistricts have not been able to obtain enough revenue to support their functions and have had to receive grants and programmatic support from the central government [13]. In addition, many health subdistricts experience frequent stock-outages and broken equipment [14]. Lack of supplies and equipment also impacts healthcare provision and safe practices, regardless of attendance by skilled professionals. When stock-outs occur, the burden to provide needed supplies falls on the patients and their families [14]. In order to address the frequent lack of supplies for safe births, Uganda's Ministry of Health, with the support from the WHO, launched Maama Kits (also called MAMA kits), to provide a low-cost "clean and safe child delivery care kit" for mothers to reduce the incidence of infection and sepsis after delivery [15]. MAMA Kits are packages that women or their family members

can purchase before they give birth. These kits contain the basic supplies for a safe delivery, with items such as gloves, razors, cord ligature, soap, and sanitary pads [15]. Though Uganda abolished health facilities user fees in government and other public health facilities, many lack the basic supplies and consumables necessary for a safe and clean delivery [13,16].

As patients and their families face additional financial burdens for essential healthcare services, monetary remittances can serve as an additional source of financial support. The additional income from any source of monetary remittances can reduce barriers to healthcare utilization, alleviate household financial constraints for food, clothing, and other needs, or allow families to invest in areas outside of food, such as land and housing [17]. Studies on the impacts of remittances, particularly in Latin America, have demonstrated positive impacts on household-level standards of living, use of preventative healthcare, and education of children in remittance-receiving households. Studies have also demonstrated positive impacts of remittances on community-level infrastructure and businesses [17–19]. A 2007 study by Acosta et al.'s examined the development impacts of remittances in eleven Latin American countries using household surveys and concluded that remittances reduced poverty in recipient households [20]. This paper intends to explore whether there are similar impacts of remittances on households in Uganda.

This study examines the association between household receipt of remittances on the prevalence of births attended by skilled providers as well as the association of remittances with access to MAMA Kits. In addition, this study intends to determine whether there is a dose-response or threshold effect of remittances received on maternal healthcare utilization. The Uganda National Panel Survey (2019/2020) was used for a secondary analysis of household-level demographic, health, and economic data (n = 643). The underlying assumption for this study was that remittances are important sources of income to households that receive them, and that the additional income from remittances will impact household use of healthcare services (see Fig 1). The objective of this study was to examine the odds of skilled delivery among households with women of reproductive age who receive remittances compared with those who do not. The hypothesis for the cross-sectional study was that households that receive remittances will have a higher prevalence of use of skilled delivery than households that did not receive remittances. The hypothesis for the dose-response analysis was that as the amount of remittances household receive increases, so does the prevalence of use of skilled delivery.

## Methods

### Ethics statement

A proposal for the research, data sources, and survey instruments was submitted to the University of North Carolina, Chapel Hill institutional review board (IRB) (UNC IRB Study No. 23–1898). An IRB exemption was approved due to the use of secondary data that is publicly available. No additional ethical approval was necessary. In accordance with IRB instructions, all data was stored and analyzed on a password protected computer with endpoint protection, and only the principal investigator (CD) had access to the raw data. In accordance with Uganda's Statistics Act of 1998, approval for conducting surveys and censuses is provided by the Board of the Uganda Bureau of Statistics. They review all censuses and surveys, as part of the National Statistical System, ensuring that data collection follows specified rules and regulations regarding financing, survey methodology and analysis, data security, and data collection timelines [21,22].

### Conceptual framework

The conceptual model in Fig 1 indicates broadly relevant individual, household and community/environmental characteristics, observed and unobserved, and the directions of their effects, first, on receipt of remittances, and then on the health outcome variables of interest. Household characteristics include variables that may bias the association between receipt of remittances and utilization of skilled delivery [17]. In this conceptual model, based on available data for Uganda, remittances refer to any monetary remittances households received within the last 12 months before the survey. In the unobservable category "remittance-sending individual" refers to any individual who may have previously migrated more than a

PLOS Global Public Health

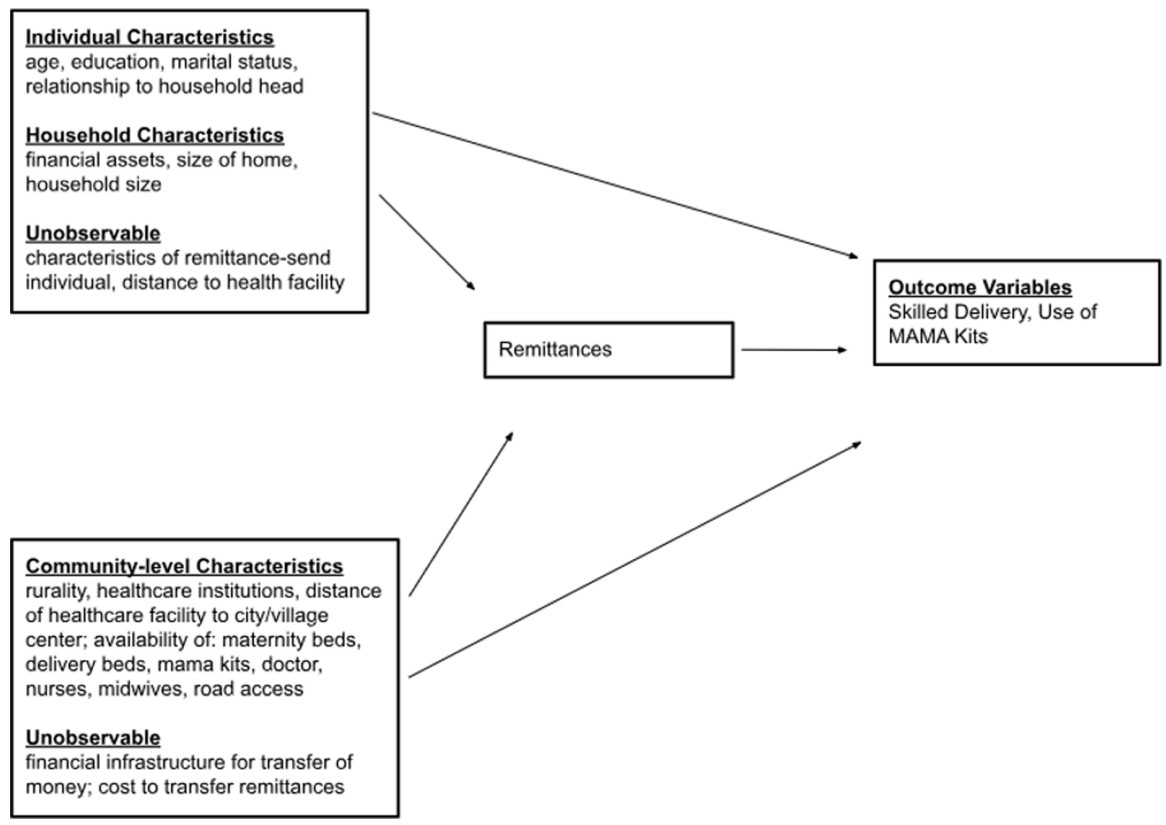

**Fig 1. Conceptual framework of relationships between remittances and other individual, household and community-level factors and outcome measures, with potential confounders and unobservable variables.**

year prior to data collection or any individual who sends remittances to the household. "Remittance-sending individual" is considered an unobservable variable because the data available in the UNPS on absent family members does not provide geographical information on their new residence location, whether another country, different district in Uganda, or even just in the same village, district, or enumeration area of the survey.

Possible confounders of the relationship between remittances and the outcomes of interest include the number of children and adults in the household and receipt of government assistance, such as cash assistance or other safety net aid. Gender, marital status and education of the household head, and economic factors including employment and dwelling quality (see S1 Table) may also have an impact on outcomes. The UNPS gathers data on members of the household who have been absent for up to 12 months but does not collect data on prior migration of household members. It asks whether individuals left and provides a list of reasons, including "to look for work" and "other economic reasons."

## Data source

This analysis uses the 2019–2020 Uganda National Panel Survey (UNPS). The first Uganda National Panel Survey (UNPS) was conducted in 2009–2010. It is sponsored by the Government of Uganda and the World Bank Living Standards Measurement Survey program [23]. The UNPS was created to fill in data gaps between various national household surveys to allow the government to properly evaluate their Poverty Eradication Action Plan. The original survey included the following questionnaires: Socio-economic, Woman, Agriculture, Community, and Prices [24]. Over the years, the survey has been updated to include additional questionnaires and additional households have been added to the sample as

part of sample refresh where they replace a third of the sample [23]. The most recent UNPS includes the following questionnaires: Household, Woman, Agriculture and Livestock, Fisheries, Community, and Market. In the 2019/2020 UNPS there was also an additional module for biological data collection for hemoglobin levels and malaria testing for women and children under 5 years old [23]. The UNPS provides the Ugandan Government with annual estimates to monitor poverty and to experiment with different policy interventions as well as evaluate existing programs. These estimates support Uganda's National Development strategy on agricultural services, budgets, household income and consumption as well as tracking poverty [23].

The UNPS collects data from a nationally representative sample of households, with over-sampling of certain rural districts. Data is collected on the household, individual, land parcel, agricultural plot, and at community-levels. Each wave of the panel survey occurs in 12-month increments. During each wave, households were visited once every six months between March 2019 and February 2020. The 2019/2020 UNPS collected data from 3,098 households and on 16,054 individuals [25]. The sample used in the analysis here on skilled delivery and use of MAMA kits include 643 households, each having a woman who reported a recent birth (within the previous three years of the survey interview). Births in the last three years were included in the analysis to capture births that occurred in the year prior to the first and last household survey interview (2018–2020). In this analysis, data from the Woman, the Household, and the Community Questionnaires were used to analyze the impact of remittances on skilled delivery.

## Variables and measurement

**Outcomes of interest.** The outcomes of interest are skilled delivery and skilled delivery with the use of a MAMA Kit (the Uganda Ministry of Health refers to these as Maama Kits, but the UNPS uses MAMA Kit). The outcomes are measured at the individual-level for women of reproductive age in each household and grouped according to remittance status (households receiving remittances and those not receiving remittances). Two questions on delivery assistance and provision of MAMA kits capture skilled delivery: Who assisted with the delivery of (NAME)?; and Were you provided with a MAMA kit?. There were seven possible responses for the delivery assistance question, with options for different healthcare personnel. The MAMA kit question had four possible responses for receipt of MAMA kits: free of charge; with a fee; no receipt of a kit; or unknown. Skilled delivery was defined using the World Health Organization definition: "Skilled delivery is defined as a birth attended by a 'skilled health personnel', as referenced by SDG indicator 3.1.2---competent maternal and newborn health (MNH) professionals educated, trained and regulated to national and international standards. They are competent to: (i) provide and promote evidence-based, human-rights based, quality, socio-culturally sensitive, and dignified care to women and newborns; (ii) facilitate physiological processes during labour and delivery to ensure a clean and positive childbirth experience; and (iii) identify and manage or refer women and/or newborns with complications. In addition, as part of an integrated team of MNH professionals (including midwives, nurses, obstetricians, pediatricians and anesthesiologists), they perform all signal functions of emergency maternal and newborn care to optimize the health and well-being of women and newborns. Within an enabling environment, midwives trained to International Confederation of Midwives (ICM) standards can provide nearly all of the essential care needed for women and newborns" [9]. A birth with a doctor or nurse/midwife is considered skilled delivery and coded as a binary variable (Yes/No). An additional variable for skilled delivery with use of MAMA kit was also coded as a binary (Yes/No) only if respondent reported being assisted by a doctor or nurse midwife during birth and use of a MAMA kit.

MAMA kits refer to clean delivery kits, either provided by the government or non-governmental organizations, that women can obtain at the time of delivery in a facility. MAMA kits are distributed through Uganda's drug distribution system to community health facilities and traditional birth attendants. Partner organizations also support education and sensitization about clean delivery and distribute kits directly to expectant mothers [15]. These kits include items such as gauze, cotton wool, razor blades, antibiotic ointment, disposable gloves, syringes, soap, and umbilical cord ties [26]. In Uganda

specifically, the Ministry of Health created the MAMA Kits to include sterile gloves, plastic sheets, cord ligature for the umbilicus, razor blades, tetracycline (antibiotic), cotton, soap, and sanitary pads [15]. Finally, one question in the survey addressed location of delivery, with nine categorical response options, which were collapsed into three: home, public, and private sector facilities. The S1 Table shows the survey questions used to create the variables in this study.

**Exposure.** The exposure, or key explanatory variable, in this study is the receipt of remittances. Information on remittances is collected in the household module of the UNPS, involving questions on remittances received locally and abroad, and the amount (S1 Table). One binary (Yes/No) question in the survey was "Has the household received any income (in cash or in kind) from [...] in the past 12 months?" There were 15 possible income sources in a list, two being "remittances and assistance received locally" and "remittances and assistance from abroad". The variable for amount of remittances is a continuous measure generated from the question following the income source question: "Amount received during the past 12 months. If amount was in kind, give the estimated cash value." It should be noted that remittances can be received from anyone: members and non-members of the household. The UNPS survey also does not specify who sends remittances to the household, whether they are migrant household members or nonmembers, relative, or friends. The primary exposure variables will be based on the estimated total value of both in-cash and in-kind remittances as reported. A binary variable will also be included for remittances received (Yes/No). Separately, the total amount of remittances a household received was measured as a continuous variable. The binary variable accounts for households that reported receipt of remittances but did not report the amounts of remittances received. About 40% of the households that received remittances did not report the amount.

The total amount of remittances may have different effects on households of varying wealth status (i.e., if a wealthy household receives the same amount of remittances as a low-income household, the impact of the remittances in the low-income household may be greater than in the wealthy household). To account for differences in household wealth, a covariate for the number of household assets was used as a proxy for household wealth. Some studies account for differences in wealth using a wealth index, consumption quintiles, or asset quintiles [17,27]. The UNPS survey did not collect data on incomes from wages, safety net grants, or farming, but did collect on income from investments, rents and royalties, and other transfers. Thus, income data are incomplete, and using the latter amounts would bias the sample towards high-income households and be of little use. An attempt was made to collect total household expenditures data, which required households to estimate various categories of expenses ranging from the last 30 days (food, transportation, rent, etc.) to the last 12 months (viz., durable and consumable goods). These measures are susceptible to recall errors and were sometimes missing. Therefore, the number of assets reported was taken to be the most reliable proxy for the economic situation of the household.

**Covariates.** There are several factors available from the UNPS that could affect access to and use of skilled delivery at birth and use of MAMA kits. These are needed in a multivariate quantitative study of the factors affecting the health outcome variables, to control for possible confounding due to the individual, household- and community-level characteristics that may influence household use of skilled delivery. These covariates include the individual variables indicating whether the woman is head of household, age, marital status, education, and occupation; and the household variables comprising number of household members (number of children 0–14 years old, number of adults 15+years old), number of rooms in the home, number of assets, and whether urban or rural.

Covariates were selected partly drawing upon limited previous studies on the effects of remittances on maternal and child health outcomes, primarily in Latin America (Ecuador, Guatemala, Mexico) and Southeast Asia [17,18]. Studies of other financial assistance programs in other countries use similar kinds of variables [28–30]. In the conceptual model the individual, household and community variables in the covariate section (Fig 1) are informed by Lindstrom et al.'s "enabling resources" and López-Cevallos et al. [17,18]. Some of these variables, or enabling resources, can influence where, when, and who decides to send remittances back to households. The inclusion of both recent remittances and all the covariates that could directly affect skilled delivery or healthcare utilization potentially weakens the effects of most of the variables,

as remittances could have been sent over a period of years before the 12-month period examined in the survey, either regularly or irregularly. This may not account for the cumulative effects of remittances on households over time.

Thus, whether remittances are sent will depend largely on the economic situation of the remittance-sending person where he/she lives, which has no causal linkage with the current characteristics of the household head or the household. As mentioned previously, the UNPS household module does not explicitly ask about household migration, nor the remittance-sending individual, the destination or current place of residence, nor his/her/their economic situation. The UNPS only has limited questions about any absent household members and their main reason for being absent, which can be used as a rough proxy for migration. Respondents are asked about household members absent for 12 months, given 14 options (S1 Table). Tests of multicollinearity among the explanatory variables used in the multivariate statistical model are reported in the discussion below.

**Analytic approach.** UNPS 2019–2020 data were accessed from the World Bank Microdata website [23]. Fifteen household, one woman, and five community datasets were cleaned and merged together to create one skilled delivery dataset with all outcomes, exposures, and covariates. This dataset was restricted to households with women of reproductive age (ages 15–49) who had a birth in the last three years prior to the interview. Observations without reported place of birth or date of last birth were deleted from the dataset (n = 1157). Each observation was grouped according to a household identification code. For households with more than one woman of reproductive age giving birth in the three-year period, one woman was randomly selected to include in the analysis. This was done to avoid correlation bias across individual, household, and community-level variables.

Categorical and binary variables were created for outcome variables, exposure variables and covariates. Distributions of observations for each variable in the model were examined and outliers identified. Outliers for remittances and number of rooms were removed from the dataset (n = 9). Descriptive statistics were then calculated to determine the balance between exposure groups and whether there were statistically significant differences between the two groups. This information informed further testing of covariates impacts on the relationship between skilled delivery and receipt of remittances prior to multivariate analysis.

Estimates of prevalence, prevalence differences, and prevalence odds ratios of skilled delivery under exposure to remittances were calculated from data on frequencies and from regression models. Unadjusted or crude prevalence of skilled delivery by remittance status (yes or no) was calculated. Prevalence differences were estimated using linear regression models and prevalence odds ratios were estimated using logit regression models. Household-level covariates were added to the model first. Due to multi-collinearity of most community-level characteristics (communities either had or did not have most of the forms of community infrastructure available), community-level variables were explored by adding variables individually to the models with individual and household characteristics already in the model. Table 5 and Table 8 include all the coefficients for each community-level variable. There were four iterations of this analysis, the first using the binary remittance exposure variable (yes, no), and the second with the continuous remittance variable to assess impacts on skilled delivery. The third and fourth iterations used the same binary and continuous exposures, with the outcome variable being skilled delivery with use of MAMA kits. Equation 1 provides the full model for this analysis. All analyses was completed in R Studio version 2022.07.1+544.

P-values of the coefficients in logistic regressions were calculated using Wald Tests (likelihood ratio tests). An $\alpha = 0.05$ was used, as is customary, for the threshold to determine statistically significant effects of variables on skilled delivery, including of remittances.

The equation below is the general framework for regression used in this analysis [31,32].

$$Y_i = \beta_o + \beta_1 R_{ik} + \beta_2 X1_j + \beta_3 X2_i + \mu_j + \varepsilon_{ij} \qquad (1)$$

Where:

$Y_ik$ = skilled delivery OR skilled delivery with MAMA kit, of $i$th household, in $k$th community;

$R_{ik}$ = remittances received by $i$th household in $k$th community (total amount of remittances; otherwise, binary where 1 = remittance received);

$Xi_k$ = household characteristics of $i$th household in $k$th community;

$X_{jk}$ = community characteristics in $k$th community;

$\mu_k$ = community random error;

$\varepsilon_{ik}$ = household random error;

$\beta_o$ = constant term;

$\beta_1$ = regression coefficient for remittances;

$\beta_2$ = regression coefficient for community characteristics;

$\beta_3$ = regression coefficient for household characteristics

For the dose-response curves, the adjusted logistic regression models for skilled delivery fitted to the data were used to predict the probabilities of skilled delivery. The dose in this scenario was the amount of remittances the household reported receiving in the last 12 months. The predicted probabilities of skilled delivery for a household are estimated by inserting all the actual values for the household in the statistical equation estimated for all households. These predicted probabilities of skilled delivery outcomes were then plotted against the dose of remittance to visualize the relationship between exposure and outcome, demonstrating how the probability of skilled delivery and skilled delivery with MAMA Kits changes with increases in remittances. This analysis was conducted only on households that reported amounts of remittances, with the amounts in Ugandan Shillings (UGX) converted to United States Dollars (USD). Loess, or locally weighted least square regression, curves were fitted to the plots of predicted probabilities because their relationship with the amount of remittances is non-linear with fluctuations in the data. The loess is a modeling method that allows for more flexibility than linear regression models in depicting the relationship between independent and dependent variables [33,34].

## Results

### Sample characteristics

There were 643 women of reproductive age who had a birth within the last three years of the survey and who reported on the location of their last birth. The mean age of women in the sample was 29 years, with 48% aged 25–34. Approximately half of the women had incomplete primary school education (49%), 15% completed primary school, and 27% had some secondary education or above. Most women were married (85%) either monogamously or polygamously, and most lived in a rural area (80%). For their last birth, women usually gave birth in government or private hospital or clinic (82%) while 18% gave birth elsewhere, usually in their home or someone else's home. Among women that reported household expenditures, median annual consumption was 7,122,000 UGX ($1867). In this sample, 17% of women reported being the head of household. In addition, the mean number of children in each household was 3.6 and the mean number of adults was 2.9. Location, number of adults in the household, and number of households assets differ significantly between households with women that utilized skilled delivery versus those that did not. See Table 1 below for the complete socio-demographic information. Tests of collinearity demonstrated that the number of rooms in a household is positively correlated with the number of adults in the household. Additionally, access to a government facility is also positively correlated with road access.

Twenty-seven percent of the women in the sample lived in households that received remittances (Table 1), which includes remittances received domestically and from abroad, in-cash and/or in-kind. Most households in the sample received remittances from domestic sources (97%). The mean amount of remittances received from abroad was on average 1.7 times larger than remittances received from internal sources. Skilled delivery, or having a doctor, nurse, or midwife present during delivery, occurred in 77% of women's last births. Comparatively, 51% of women had a skilled delivery with use of a MAMA kit during their last birth (Table 1).

**Table 1. Socio-demographic characteristics of women in study sample (means except as noted).**

| | Skilled Delivery | | | |
| --- | --- | --- | --- | --- |
| | Overall (n = 643) | No (n = 145) | Yes (n = 498) | p-value[3] |
| Individual Characteristics | | | | |
| Age[2] | 29 (24, 34) | 29 (24, 34) | 29 (24, 34) | 0.6 |
| Woman/Respondent Household Head[1] | | | | 0.093 |
| Yes | 108 (17%) | 31 (21%) | 77 (15%) | |
| No | 535 (83%) | 114 (79%) | 421 (85%) | |
| Education[1] | | | | |
| Some Primary | 315 (49%) | 82 (57%) | 233 (47%) | |
| Completed Primary | 93 (14%) | 20 (14%) | 73 (15%) | |
| Some Secondary | 121 (19%) | 18 (12%) | 103 (21%) | |
| Completed Secondary or Above | 56 (8.7%) | 1 (0.7%) | 55 (11%) | |
| Not reported/DK | 58 (9.0%) | 24 (16.7%) | 34 (6.8%) | |
| Married[1] | | | | 0.2 |
| Yes | 548 (85%) | 128 (88%) | 420 (84%) | |
| No | 95 (15%) | 17 (12%) | 78 (16%) | |
| Skilled Delivery with MAMA Kit[1] | | | | <0.001 |
| Yes | 326 (51%) | 0 (0%) | 326 (65%) | |
| No | 317 (49%) | 145 (100%) | 172 (35%) | |
| Household Characteristics | | | | |
| Location[1] | | | | 0.003 |
| Urban | 126 (20%) | 16 (11%) | 110 (22%) | |
| Rural | 517 (80%) | 129 (89%) | 388 (78%) | |
| Number of Adults[2] | 2.00 (2.00, 3.00) | 2.00 (2.00, 3.00) | 2.00 (2.00, 4.00) | 0.032 |
| Number of Children[2] | 3.00 (2.00, 5.00) | 3.00 (3.00, 5.00) | 3.00 (2.00, 5.00) | 0.4 |
| Number of Assets[2] | 6.0 (4.0, 8.0) | 5.0 (3.0, 7.0) | 6.0 (4.0, 8.0) | 0.01 |
| Received Remittance[1] | | | | 0.5 |
| Yes | 175 (27%) | 36 (25%) | 139 (28%) | |
| No | 468 (73%) | 109 (75%) | 359 (72%) | |
| Community-level Characteristics | | | | |
| Government Health Facility Available*[1] | | | | |
| Yes | 426 (99%) | 103 (100%) | 323 (98%) | 0.6 |
| No | 5 (1.2%) | 0 (0%) | 5 (1.5%) | |
| Private facility available*[1] | | | | |
| Yes | – | – | – | |
| No | 430 (100%) | 103 (100%) | 327 (100%) | |
| Road Access*[1] | | | | 0.4 |
| Yes | 425 (94%) | 100 (93%) | 325 (95%) | |
| No | 26 (5.8%) | 8 (7.4%) | 18 (5.2%) | |

[1] n (%); [2] Median (IQR); [3] Pearson's Chi-squared test; Wilcoxon rank sum test; Fischer's exact test *Missing Data.

Crude prevalence of skilled delivery and the frequencies of socio-demographic characteristics by receipt of remittances (or "exposure") group are listed in Table 2. When examining unadjusted prevalence of skilled delivery and skilled delivery with use of a MAMA kit, there were no differences according to household receipt of remittances. In the no remittance group, 77% of women had a skilled delivery compared to 79% in the remittance group. Results were similar for skilled

**Table 2. Frequencies: Skilled delivery & skilled delivery with MAMA kit by exposure to remittances.**

| | Outcome Measure (n = 1,387) | Crude Prevalence Exposed (Received Remittance) | Crude Prevalence Unexposed (No Remittance) | Crude Prevalence Difference |
|---|---|---|---|---|
| Binary Remittance Exposure | Skilled Delivery | 0.79 | 0.77 | 0.03 |
| | Skilled Delivery + MAMA Kit | 0.52 | 0.50 | 0.02 |
| Continuous Remittance Exposure (Sum of remittances) | Skilled Delivery | 0.79 | 0.77 | 0.03 |
| | Skilled Delivery + MAMA Kit | 0.52 | 0.50 | 0.02 |

delivery with use of a MAMA kit (50% in no remittance vs. 52% if received remittances); about half of women in both exposure groups used a MAMA kit. On a population level, each percentage point difference between the exposed and unexposed group accounts for approximately 17,111 births per year.

When comparing socio-demographic characteristics between exposure groups only two covariates tested had statistically significant differences between households that received remittances and those that did not. Marital status (married vs. unmarried) between the groups differed significantly (p < 0.001), with married women making up 90% of the unexposed group and 71% of the exposed group. The distribution of women who were the household head was also statistically significantly different between exposure groups (p < 0.001). Thirteen percent of women in the unexposed group were heads of their households compared with 28% in the exposed group. There was also a statistically significant difference in location (14% of women in remittance receiving households live in urban setting vs. 22% in non-remittance receiving households) and in the number of adults in the household (median of 3 adults in remittance receiving households vs. 2 in non-remittance receiving households).

## Regression models

Tables 3 and 4 presents the results of the logistic regression models of the relationship between skilled delivery and skilled delivery with use of MAMA kits with remittances and the other household and community-level covariates. Estimates of odds ratios (OR) with their corresponding confidence intervals are included in the table. A woman with a primary school education and rurality had statistically significant relationships with skilled delivery and skilled delivery with use of a MAMA kit. In addition, there was a statistically significant relationship between skilled delivery with MAMA kits and a woman being the household head. Having at least completed primary school education was strongly positively associated

**Table 3. Regression model summary outputs: Skilled delivery (Binary exposure).**

| Covariate | OR Estimate | Lower 95% CI | Upper 95% CI |
|---|---|---|---|
| Remittance | 1.21 | 0.76 | 1.91 |
| Completed Primary Education or above | 2.03*** | 1.33 | 3.10 |
| Age | 0.99 | 0.96 | 1.03 |
| Married | 1.00 | 0.52 | 1.91 |
| # Children | 0.95 | 0.84 | 1.08 |
| Urban | 1.96** | 1.09 | 3.54 |
| Household Head | 0.77 | 0.45 | 1.32 |
| # Rooms | 1.28 | 0.98 | 1.67 |
| # Adults | 1.00 | 0.84 | 1.20 |

All continuous predictors are mean-centered and scaled by 1 standard deviation. The outcome variable is in its original units. *** p < 0.001; ** p < 0.01; * p < 0.05.

Pseudo $R^2$ = 0.05.

**Table 4. Regression model summary outputs: Skilled delivery with MAMA kit (Binary exposure.).**

| Covariate | OR Estimate | Lower 95% CI | Upper 95% CI |
|---|---|---|---|
| Remittance | 1.22 | 0.83 | 1.78 |
| Completed Primary Education or above | 1.91*** | 1.37 | 2.67 |
| Age | 1.00 | 0.98 | 1.03 |
| Married | 1.02 | 0.60 | 1.73 |
| # Children | 0.90 | 0.82 | 1.00 |
| Urban | 1.78** | 1.15 | 2.75 |
| Household Head | 0.55* | 0.34 | 0.88 |
| # Rooms | 0.98 | 0.81 | 1.18 |
| # Adults | 1.06 | 0.91 | 1.22 |

All continuous predictors are mean-centered and scaled by 1 standard deviation. The outcome variable is in its original units. *** $p < 0.001$; ** $p < 0.01$; * $p < 0.05$.

Pseudo $R^2 = 0.05$.

with skilled delivery (OR 2.03 (CI 1.33, 3.10)) and with skilled delivery with a MAMA kit (OR 2.02 (CI 1.32, 3.08)). Living in an urban setting was positively associated with skilled delivery (OR 1.96 (CI 1.09, 3.54)) and use of MAMA kits (OR 1.78 (CI 1.06, 3.41). If the head of household was the female respondent, this was associated with lower odds of skilled delivery with MAMA kits under binary and continuous exposure to remittances (ORs 0.55 (CI0.34, 0.88) and 0.47 (CI 0.29, 0.78), respectively). There are no statistically significant relationships between receipt of remittances and utilization of skilled delivery or skilled delivery with use of MAMA kits.

Community-level variables were added to the logistic regression models that were fully adjusted for household-level characteristics to determine if there were any discernable community-level affects once all the other variables were taken into account. The community-level characteristics were added separately, one by one, to avoid possible multicollinearity. The results of these models are in Table 5 below. None of the community-level characteristics had statistically significant impacts on the odds ratios of skilled delivery or skilled delivery with MAMA kits. For skilled delivery with MAMA kits, the largest changes in odds ratios were for road access variables. Community-level variables had similar effects on the odds of skilled delivery, with most odds ratios ranging from 1.51 to 1.59 (excluding road access). Community-level variables also had similar effects on skilled delivery with the use of MAMA, however odds of skilled delivery were lower than the model adjusted only for household characteristics. It is important to note that community-level variables had significant and systematic missing data. The association between remittances and use of skilled delivery and skilled delivery with MAMA kits was weaker than other covariates in the model (education, woman as household head, and rurality).

Logistic regression models of skilled delivery and skilled delivery with use of MAMA kits were repeated similarly to previous models, but with amount of remittances as the exposure. Amount of remittances was measured as UGX. As Tables 6 and 7 depict, completion of primary education, living in an urban location, and the woman as the household head had statistically significant associations with skilled delivery and skilled delivery with use of MAMA kits. Primary education and urban location increased the odds of skilled delivery, whereas the woman as household head decreased the odds. There appears to be no change in odds of skilled delivery and skilled delivery with MAMA kits with increasing amounts of remittances.

Community-level variables were also added to the models with the continuous measure of the amount of remittances (Table 8). The availability of nurses in the community was the only community-level variable that demonstrated a statistically significant relationship with skilled delivery (OR 0.90, CI(0.81, 1.00). Interestingly, nurse availability was associated with lower odds of skilled delivery which is contradictory to previous literature. It is important to note, that though statistically significant, the confidence intervals include the null value. About half of the sample was missing community-level data on healthcare personnel, including nurses.

**Table 5. Adjusted odds ratios of skilled delivery and skilled delivery with MAMA kits with community-level variables added one by one (Binary remittance exposure).**

| Adjustments | Skilled Delivery | | Skilled Delivery+MAMA Kit | |
|---|---|---|---|---|
| | OR (95% CI) of community-level variable | p-value | OR (95% CI) | p-value |
| HC+ Distance of Health Facility to Trade Center | 1.07 CI: (0.90, 1.31) | 0.45 | 0.93 (0.79, 1.09) | 0.40 |
| HC+ Delivery Services | 0.78 CI: (0.41, 1.42) | 0.43 | 1.41 (0.84, 2.40) | 0.19 |
| HC+MAMA Kit Availability | 0.82 CI: (0.50, 1.35) | 0.45 | 1.15 (0.74, 1.77) | 0.53 |
| HC+ Delivery Beds | 0.67 CI: (0.35, 1.20) | 0.19 | 1.28 (0.78, 2.13) | 0.33 |
| HC+Maternity Beds | 0.98 CI: (0.94, 1.02) | 0.39 | 1.00 (0.96, 1.03) | 0.86 |
| HC+Road Access | 1.92 CI: (0.73, 4.75) | 0.17 | 1.03 (0.43, 2.43) | 0.95 |
| HC+Nurse Availability | 0.90 CI: (0.82, 1.00) | 0.05* | 0.97 (0.88, 1.06) | 0.52 |
| HC+Doctor Availability | 1.16 CI: (0.79, 1.78) | 0.48 | 1.11 (0.79, 1.54) | 0.55 |
| HC+Midwife Availability | 1.01 CI: (0.88, 1.18) | 0.89 | 0.98 (0.87, 1.11) | 0.80 |

All continuous predictors are mean-centered and scaled by 1 standard deviation. The outcome variable is in its original units. *** $p < 0.001$; ** $p < 0.01$; * $p < 0.05$.

**Table 6. Regression model summary outputs: Skilled delivery (Continuous exposure).**

| Covariate | OR Estimate | Lower 95% CI | Upper 95% CI |
|---|---|---|---|
| Remittance | 1.00 | 1.00 | 1.00 |
| Primary Education Completed | 2.02** | 1.32 | 3.07 |
| Age | 0.99 | 0.96 | 1.03 |
| Married | 0.99 | 0.52 | 1.89 |
| # Children | 0.96 | 0.84 | 1.08 |
| Urban | 1.90* | 1.06 | 3.42 |
| Household Head | 0.73 | 0.43 | 1.26 |
| # Rooms | 1.26 | 0.96 | 1.64 |
| # Adults | 1.00 | 0.84 | 1.20 |

All continuous predictors are mean-centered and scaled by 1 standard deviation. The outcome variable is in its original units. *** $p < 0.001$; ** $p < 0.01$; * $p < 0.05$.

Pseudo $R^2 = 0.05$.

The regression models demonstrate that most of the community-level covariates had no associations with skilled delivery and skilled delivery with MAMA kits. These covariates would have little effect on the relationship between exposure to remittance and the skilled delivery outcomes. Given previous literature, the education of the woman, location, and having the role of household head have significant associations with skilled delivery utilization. In all regression models, there were no statistically significant differences in skilled delivery outcomes between the two remittance exposure groups. The association of these variables with skilled delivery and skilled delivery with MAMA kits were stronger than exposure to remittances.

## Dose-response relationships

The dose-response relationships between the amount of remittances received in a year (in USD), skilled delivery and skilled delivery with MAMA kits is depicted in Fig 2 and Fig 3. Both plots demonstrated the wide variability in coverage of skilled delivery among households that receive between $0 and $100. Few households received greater than

**Table 7. Regression model summary outputs: Skilled delivery with MAMA kit (Continuous exposure).**

| Covariate | OR Estimate | Lower 95% CI | Upper 95% CI |
|---|---|---|---|
| Remittance | 1.00* | 1.00 | 1.00 |
| Primary Education Completed | 1.89*** | 1.35 | 2.64 |
| Age | 1.00 | 0.98 | 1.03 |
| Married | 1.06 | 0.63 | 1.78 |
| # Children | 0.91 | 0.82 | 1.00 |
| Urban | 1.74* | 1.13 | 2.68 |
| Household Head | 0.47** | 0.29 | 0.78 |
| # Rooms | 0.95 | 0.78 | 1.16 |
| # Adults | 1.05 | 0.91 | 1.23 |

All continuous predictors are mean-centered and scaled by 1 standard deviation. The outcome variable is in its original units. *** $p < 0.001$; ** $p < 0.01$; * $p < 0.05$.

Pseudo $R^2 = 0.06$.

**Table 8. Adjusted odds ratios of skilled delivery and skilled delivery with MAMA kits with community-level variables added one by one (Continuous remittance exposure).**

| Adjustments | Skilled Delivery | | Skilled Delivery + MAMA Kit | |
|---|---|---|---|---|
| | OR (95% CI) | p-value | OR (95% CI) | p-value |
| HC+ Distance of Health Facility to Trade Center | 1.07 (0.90, 1.30) | 0.49 | 0.94 (0.79, 1.10) | 0.46 |
| HC+ Delivery Services | 0.79 (0.41, 1.43) | 0.45 | 1.45 (0.86, 2.46) | 0.17 |
| HC+MAMA Kit Availability | 0.83 (0.49, 1.35) | 0.45 | 1.13 (0.74, 1.75) | 0.57 |
| HC+ Delivery Beds | 0.68 (0.36, 1.23) | 0.22 | 1.30 (0.79, 2.16) | 0.31 |
| HC+Maternity Beds | 0.98 (0.94, 1.02) | 0.36 | 1.00 (0.96, 1.03) | 0.81 |
| HC+Road Access | 1.92 (0.73, 4.77) | 0.17 | 1.09 (0.45, 2.60) | 0.85 |
| HC+Nurse Availability | 0.90 (0.81, 1.00) | 0.04* | 0.96 (0.88, 1.06) | 0.44 |
| HC+Doctor Availability | 1.14 (0.78, 1.76) | 0.51 | 1.10 (0.79, 1.53) | 0.56 |
| HC+Midwife Availability | 0.99 (0.85, 1.17) | 0.92 | 0.97 (0.85, 1.09) | 0.58 |

All continuous predictors are mean-centered and scaled by 1 standard deviation. The outcome variable is in its original units. *** $p < 0.001$; ** $p < 0.01$; * $p < 0.05$.

$150. Loess curves fitted to the scatterplots of predicted probabilities of skilled delivery and skilled delivery with MAMA kits demonstrated a decrease in skilled delivery at remittance amounts less than $75. At remittances above $75 the probability of skilled delivery at birth increased and then plateaued at approximately $200. The loess curve remained between 0.70 and 0.85 probabilities of skilled delivery, reflective of the high coverage of skilled delivery in Uganda. At approximately $250, coverage of skilled delivery began to decrease. It is important to note that there were less than 10 households that report receiving remittances over $250. The confidence interval bands widened as the amount of remittances increase, show the lack of precision in the loess curves fit with fewer households reporting remittance amounts greater than $250.

The dose-response curve for skilled delivery with use of MAMA kits demonstrated little change in the probability of skilled delivery between $0 and $100 (Fig 3). For remittance amounts greater than $100 USD the probability of skilled delivery at birth increased through $500. At $500 the loess curve plateaued and then decreased as remittance amounts increased to $800. Note that most households reported remittances for amounts less than $200. The widened confidence interval bands show the lack of precision in estimates with fewer reports of remittances in amounts greater than $200.

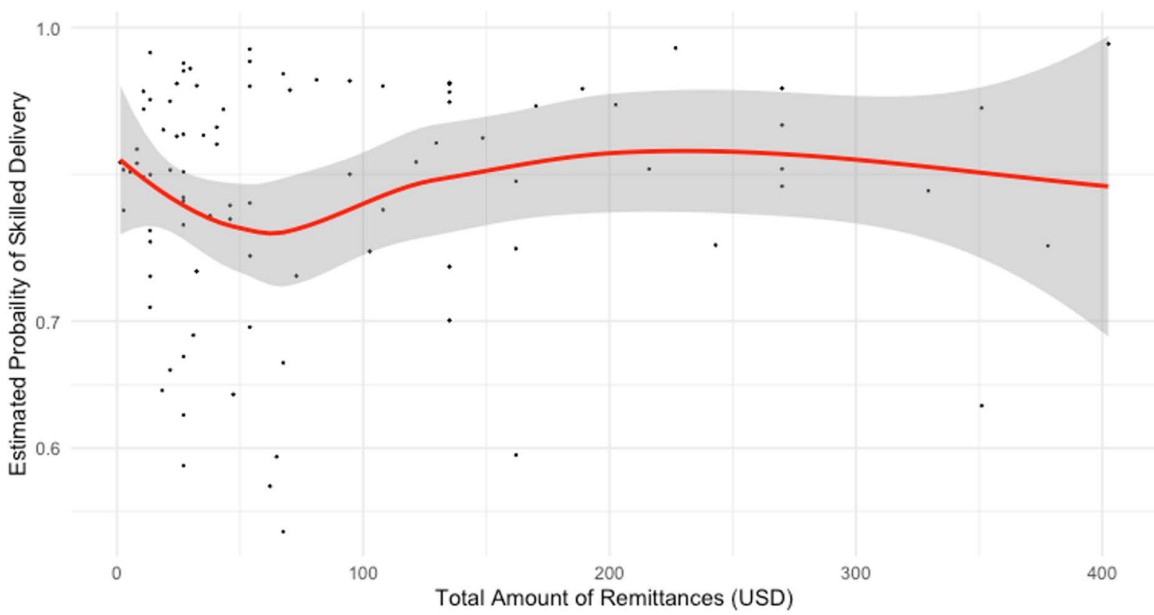

**Fig 2. Relationship of amount of remittances and estimated probability of skilled delivery (adjusted model) n =91.**

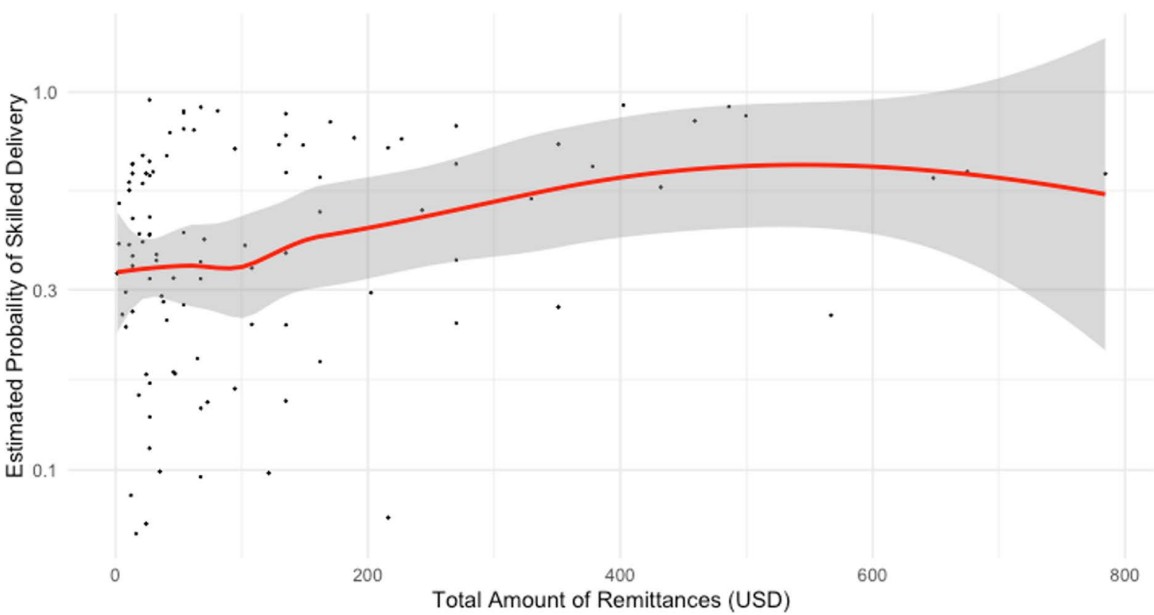

**Fig 3. Relationship of amount of remittances and estimated probability of skilled delivery with MAMA Kit (adjusted model) n =105.**

## Discussion

This study provides additional knowledge on the impacts of remittances on skilled birth attendance and usage of MAMA kits in Uganda. In addition, these findings showed that there may be more significant factors, other than financial, that affected skilled delivery and use of MAMA kits. The results of this study indicated that there is no statistically significant relationship

between the receipt of remittances in Uganda and the odds of skilled delivery and use of MAMA kits among women of reproductive age with a recent birth. Though the dose response curves showed changes in the probability of skilled delivery and skilled delivery with MAMA kits as remittances increased, they did not demonstrate a significant threshold effect.

The inconsistency between the results of the logistic regressions and the dose-response curves may also be due to the small sample size. Additionally, logistic regression analysis with the binary exposure of remittances may not be sufficient in revealing the effects of remittances on utilization of skilled delivery, particularly in the absence of data on total household income and expenditures. Among the sample of 643 women, only 175 women were in households that received remittances. The small sample size may make it difficult to detect any significant relationships between remittances and skilled delivery in the regression analysis. Within the dose-response curves, the confidence interval bands widened with fewer data points in the plots, demonstrating the uncertainty of the predicted probabilities. In further studies, examining a larger sample of women of reproductive age may reduce the uncertainty of the results.

Uganda has strict regulations regarding foreign exchange transactions and international exchange of funds. All international financial transactions need to occur through the Bank of Uganda or through a person or business with express permission and license to carry out those transactions [35]. In 2015, several advancements were made in international mobile money transfers. Previous difficulties regarding international money transfer were related to a lack of clarity around regulations of cross-border transfers and formal financial infrastructure. However, policy changes and development of new financial products have allowed for more cross-border transfers that work within the confines of Uganda's Foreign Exchange Act (2004), the Bank of Uganda and other Ugandan financial institutions. These developments allowed the expansion of mobile money use and other mobile money transfer companies like Google Pay and Apple Pay among others. There are also formal and informal methods of mobile money transfer to individuals who may not use banks or have access to virtual banking. However, these different methods require foreign exchange fees, cellular roaming charges, and/or other administrative charges [35–37].

Traditional methods of remittance transfer, such as wire transfer and mobile money, were available to individuals surveyed in the 2019/2020 UNPS. Households that did receive international remittances tended to receive larger amounts (average of $93 more). The disparity in sources of remittances and the difference in amounts received may be associated with the costs of international remittances. Those with more money may have been able to afford the fees associated with sending money internationally.

The literature on the impact of remittances on health outcomes, particularly among households with migrant members, have mixed results. Studies conducted in low- and middle-income settings either found positive effects of remittances on health outcomes or no effects [18,38–43]. Yabiku et al. and Siriwardhana et al. found that the effects of remittances on family health and well-being was dependent on the success of migrant family members, conditions of the household pre- and post-migration, and support mechanisms [44,45]. Though most literature demonstrates a positive relationship between remittances and family health outcomes, it is clear that this is a nuanced issue.

The small size of this sample may have led to imprecise results. The small sample size was due to a high level of missing data. To create the most complete set of variables, the sample was restricted to women with non-missing data on the exposure, outcomes, and key socio-demographic characteristics, though missingness still persisted for several community-level covariates. This may have led to missing data bias for missing covariates. The missingness in the study could be due to respondents' unwillingness to answer certain questions or misunderstanding of questions asked by interviewers. Either of these issues could also lend themselves to systematic error in data collection and imprecise outcome measures. Respondents were asked to recollect the amount of remittances received, both in-cash and in-kind, over the course of the previous 12 months. Instances where households reported receipt of remittances but did not report the amount contributed to possible missing data bias as well as measurement error. The use of number of assets as a measure of socioeconomic status as opposed to household income is also a limitation to this study. However, due to the lack of reliable self-reported income in the UNPS dataset, assets served as more tangible markers of wealth within a household.

Additionally, unobserved variables such as the distance to healthcare facilities, the availability of providers and resources at health facilities, transportation distances or costs, conditions of the roads, and social programs may have led to unmeasured confounding. Women were also asked to report on births that occurred during three years prior. Given Uganda's push for national policies to promote skilled birth attendance, it is possible that social desirability bias may have impacted women's responses. In addition, potential bias may have been introduced in the analysis by using a broad inclusion criterion for recent births within three years of data collection. The effects of remittances may have been inflated by including births that occurred more than 12 months before the survey, given that exposure to remittances was assessed over the previous 12-months.

The cross-sectional nature of the Uganda National Panel Survey (UNPS) may not capture the effects of national policy changes or the fluctuations in coverage of skilled delivery over time. Though the UNPS provides useful and timely information about household demographics, health, and economics, the dataset did include significant missingness across several key variables, such as household income and community-level variables. As a result of the missingness and the study's inclusion criteria only 643 women from 3,098 households and on 16,054 individuals were included in the analysis. The quality of the data and the amount of missingness may have impacted our ability to estimate the association of our outcomes of skilled delivery at birth and use of birthing kits with receipt of remittances.

Despite these study limitations, this study does increase knowledge of remittance flows and its impacts on maternal and child health in Uganda. There are few studies that include remittances and their impacts on maternal health outcomes in Sub-Saharan Africa, despite the large numbers of labor migrants. There is also a lack of data that capture both remittance data and household health information in Sub-Saharan Africa. This study demonstrates the possibility of using existing household level surveys to estimate the relationships between external financial support and maternal health outcomes. Future studies with larger sample sizes and better data on migrants and remittances are desirable to better evaluate this relationship.

## Supporting information

**S1 Table. Description of study measures: Outcome, exposure, and covariates.**
(DOCX)

## Acknowledgments

We thank Dr. Andrew Olshan for contributing feedback and comments on the manuscript for this publication, particularly as it related to research methods and the reporting of findings.

## Author contributions

**Conceptualization:** Chelsea Ducille.

**Data curation:** Chelsea Ducille.

**Formal analysis:** Chelsea Ducille.

**Investigation:** Chelsea Ducille.

**Methodology:** Chelsea Ducille, Richard Bilsborrow.

**Project administration:** Chelsea Ducille, Kavita Singh.

**Writing – original draft:** Chelsea Ducille.

**Writing – review & editing:** Chelsea Ducille, Richard Bilsborrow, Kavita Singh, Jon Hussey, Larissa Jennings Mayo-Wilson.

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
