## [Decision Letter · Decision Letter 0]

PGPH-D-24-01223

Association of remittances with skilled delivery and child health in Uganda, 2019/2020

Dear Dr. Ducille,

Thank you for submitting your manuscript to PLOS Global Public Health. After careful consideration, we feel that it has merit but does not fully meet PLOS Global Public Health’s publication criteria as it currently stands. Therefore, we invite you to submit a revised version of the manuscript that addresses the points raised during the review process.

Please note that we have only been able to secure a single reviewer to assess your manuscript. We are issuing a decision on your manuscript at this point to prevent further delays in the evaluation of your manuscript. Please be aware that the editor who handles your revised manuscript might find it necessary to invite additional reviewers to assess this work once the revised manuscript is submitted. However, we will aim to proceed on the basis of this single review if possible. 

Could you please revise the manuscript to carefully address the concerns raised?

We look forward to receiving your revised manuscript.

Kind regards,

Steve Zimmerman, PhD

PLOS Staff Editor

Journal Requirements:

Additional Editor Comments (if provided):

Reviewers' comments:

Reviewer's Responses to Questions

**Comments to the Author**

1. Does this manuscript meet PLOS Global Public Health’s publication criteria?

Reviewer #1: Yes

2. Has the statistical analysis been performed appropriately and rigorously?

Reviewer #1: Yes

3. Have the authors made all data underlying the findings in their manuscript fully available (please refer to the Data Availability Statement at the start of the manuscript PDF file)?

Reviewer #1: Yes

4. Is the manuscript presented in an intelligible fashion and written in standard English?

Reviewer #1: No

Reviewer #1: The introduction of this manuscript contains a few ambiguous statements and presents sentences that are more suggestions than statements. The information on the UNPS is scattered around the introduction and the source of data. Since UNPS is the only source of data, it is important to describe it in detail in the introduction or under the source of data thoroughly.

Although remittances information for the past 12 months is used births during the last 3 years are used in the study. What would be the justification? doe this not introduce bias?

Not being able to include the household income in the analysis is a serious draw back since assets doe not always match the income.

The analysis is comprehensive and detailed.

The limitations of the UNPS data set should be highlighted under the limitations more since, the outcome is totally dependent on the data available and quality of data.

Line 96 - introduction to remittances does not clearly mention what the authors refer to - local or foreign

Line 110 - objectives are presented in the present tense

Line 141 - other household variables...... it is not clear what the authors intend to say here, is this a statement or a question?

Line 216 - these kits include items like... are these standard kits or custom made? the word like introduces ambiguity.

Line 246 - household and be if little use - should be corrected as "be of little use"

**Do you want your identity to be public for this peer review?** For information about this choice, including consent withdrawal, please see our Privacy Policy

Reviewer #1: No

---

## [Decision Letter · Decision Letter 1]

Association of remittances with skilled delivery in Uganda, 2019/2020

PGPH-D-24-01223R1

Dear Chelsea Ducille

We are pleased to inform you that your manuscript 'Association of remittances with skilled delivery in Uganda, 2019/2020' has been provisionally accepted for publication in PLOS Global Public Health.

Best regards,

Neha Batura

Academic Editor

Reviewer Comments (if any, and for reference):

Reviewer's Responses to Questions

**Comments to the Author**

Reviewer #1: All comments have been addressed

publication criteria?

Reviewer #1: Yes

3. Has the statistical analysis been performed appropriately and rigorously?

Reviewer #1: Yes

4. Have the authors made all data underlying the findings in their manuscript fully available (please refer to the Data Availability Statement at the start of the manuscript PDF file)?

Reviewer #1: Yes

5. Is the manuscript presented in an intelligible fashion and written in standard English?

Reviewer #1: Yes

Reviewer #1: The authors have addressed the review comments

**Do you want your identity to be public for this peer review?** For information about this choice, including consent withdrawal, please see our Privacy Policy

Reviewer #1: No
